# Industrial-Scale Decontamination Procedure Effects on the Content of Acaricides, Heavy Metals and Antioxidant Capacity of Beeswax

**DOI:** 10.3390/molecules24081518

**Published:** 2019-04-17

**Authors:** María D. Navarro-Hortal, Francisco J. Orantes-Bermejo, Cristina Sánchez-González, Alfonso Varela-López, Francesca Giampieri, Cristina Torres Fernández-Piñar, Josep Serra-Bonvehí, Tamara Y. Forbes-Hernández, Patricia Reboredo-Rodríguez, Juan Llopis, Pilar Aranda, Maurizio Battino, José L. Quiles

**Affiliations:** 1Institute of Nutrition and Food Technology “José Mataix Verdú”, Department of Physiology, Biomedical Research Center, University of Granada, Avda del Conocimiento sn., 18100 Armilla, Granada, Spain; mdnavarro@ugr.es (M.D.N.-H.); crissg@ugr.es (C.S.-G.); alvarela@ugr.es (A.V.-L.); jllopis@ugr.es (J.L.); paranda@ugr.es (P.A.); 2Apinevada Analytical Laboratory of Bee Products, Barrancos s/n, Lanjarón, 18420 Granada, Spain; director@apinevada.com (F.J.O.-B.); calidad@apinevada.com (C.T.F.-P.); 3Dipartimento di Scienze Cliniche Specialistiche ed Odontostomatologiche—Sez, Biochimica, Università Politecnica delle Marche, Ancona, 60131 Ancona, Italy; f.giampieri@staff.univpm.it (F.G.); mbattino@mta01.univpm.it (M.B.); 4Research & Development Department, Mielso, S.A., Pol. Industrial ’El Mijares’, C/. Industria 1, 12550 Almassora (Castelló), Spain; serrjosep@gmail.com; 5Nutrition and Food Science Group, Department of Analytical and Food Chemistry, CITACA, CACTI, University of Vigo, 36310 Vigo, Spain; tamaraforbe@gmail.com (T.Y.F.-H.); preboredo@uvigo.es (P.R.-R.)

**Keywords:** beeswax decontamination, pesticides, monoesters, hydrocarbons, chemical elements, flavonoids, total phenols, adulteration

## Abstract

Beeswax is useful for the beekeeping sector but also for the agro-food, pharmaceutical or cosmetics sectors. Frequently, this bee product is contaminated with pesticides reducing its utility and causing the decline in its market. This study aimed to prove the effectiveness of an industrial-scale decontamination method in removing acaricides from beeswax. Chlorfenvinphos and coumaphos decrease was higher than 90%, whereas *tau* fluvalinate decrease was only 30%. No changes were observed in the beeswax content of hydrocarbons and monoesters, whereas a decrease in the concentrations of Ca, Fe, Zn, Hg, Mn and P, and an increase in the concentrations of As and Si were found after the decontamination. Filtration reduced total phenolics, flavonoids and the antioxidant capacity of the lipophilic extract. These results demonstrate that the industrial method used was as effective as the method previously tested on a laboratory scale. The study also contributes to a better knowledge and characterization of beeswax, specially related to trace and ultra-trace elements and antioxidant capacity. Moreover, it offers the chance to further develop a method to effectively detect wax adulterations based on the chemical elements profile.

## 1. Introduction

Beeswax is a natural honeybee product endogenously secreted by specific glands of worker honeybees [1]. Beeswax is vital for the honeybee colony because it is used as building material for its comb cells where nectar and honey are stored [2]. Beeswax is an extremely complex mixture of lipophilic nature composed of esters of fatty acids, hydrocarbons, and other minor substances [1,3]. Frequently, beeswax is contaminated with residues of pesticides, such as pyrethroid and organophosphate acaricides, which are used against the mite *Varroa destructor*. This mite is a parasite of bees that is currently considered the biggest threat to apiculture [4]. The long-term application of acaricides leads to the deposition of non-volatile and lipophilic residues, which accumulate especially in beeswax. Wax is collected and reused in apiculture and in other sectors, such as food, chemical, pharmaceutical or cosmetics sectors. In the agri-food industry, beeswax is used as food additive (E901), as a glazing agent in the preparation of pastries, for the treatment of some fruits, as food supplement, and as a flavor carrier [5]. Beeswax plays a role as binder, thickener, drug carrier and release retardant in pharmaceutical preparations [6]. However, contamination devaluates recycled wax, causes health problems when using wax for food or other purposes, and affects bee colonies, therefore, contamination has led to a gradual decline in the beeswax market in recent years.

Recently, our research group has tested a wax decontamination protocol [7] at a pilot scale. This new protocol sought to avoid the use of chemical agents for the bleaching of used beeswax, since these agents altered the composition of the wax and barely managed to reduce its degree of contamination [7]. The results obtained were promising, with decreases in the concentrations of highly representative chemicals. The present study digs deeper in the above-mentioned method, but now at an industrial scale. Although we must continue developing methodologies that further reduce the level of pesticides, it is also important to elucidate if any technology reducing the presence of various toxic compounds can result in changes in the chemical composition of the beeswax. The objective of the present study was to test the effectiveness of the decontamination process at an industrial scale and its effects on wax composition. For this purpose, the composition of different samples were investigated and the characterization of beeswax before and after being subjected to decontamination was performed. In particular, the study focused on the composition of monoester and hydrocarbon waxes, the content of chemical elements, including trace and ultra-trace elements, as well as the antioxidant capacity and total phenolic and flavonoid contents of hydrophilic and lipophilic extracts of the waxes.

## 2. Results and Discussion

### 2.1. Pesticide Content

The main objective of subjecting beeswax to the filtration process was to reduce the levels of pesticides. Therefore, the negative impact of pesticides on human health and on the hives would be prevented. Three types of acaricides were evaluated in beeswax samples. The initial and final concentrations of these three pesticides are specified in Table 1.

The studied compounds were chlorfenvinphos, coumaphos and *tau* fluvalinate, all of which are the most common acaricides used by Spanish beekeepers against *Varroa destructor* [8,9,10,11]. These compounds are also used in other countries such as USA [12], France [13], Belgium [14] and Italy [15]. In the first cycle of filtration, the reduction of each compound was around 20% except for chlorfenvinphos, which the reduction exceeded 40%. Nevertheless, the percentage reduction of organophosphate chlorfenvinphos and coumaphos in the sample was greater than 90% after the second cycle of filtration. Two cycles of filtration could only eliminate 636 out of 2113 ppb of *tau* fluvalinate, which led to a 30% of reduction. These percentage reductions are in the same range as those reported in previous studies on Spanish beeswax [7], on palm oil at laboratory-scale [16] and on industrial edible oils [17] and are consistent with other studies performed on olive oil [18]. These findings highlight the relevance of the molecule type and its liposolubility in the filtration process, with the pyrethroid pesticide the most lipophilic and the least eliminated. Furthermore, the incomplete elimination of organophosphates could be explained by their competition with monoesters for adsorption sites [7].

Consistent with these results, the industrial method tested for decontamination displayed its effectiveness to remove pesticide contents in beeswax samples, especially organophosphate acaricides. Therefore, the industrial reproducibility of the decontamination method performed at laboratory scale by Bonvehi & Orantes-Bermejo [7] is proven in the present study.

### 2.2. Hydrocarbons and Monoesters Content

Hydrocarbons and monoesters are two of the main groups constituting to beeswax. Their identification and quantification were established by gas chromatography-mass spectrometry (GC-MS). The beeswax composition regarding the content of aliphatic hydrocarbons was not modified by the filtration process. The hydrocarbons H_19_-H_44_ were evaluated and there were no significant differences between the existent amounts before and after filtration, as shown in Table 2. Among these substances, H_27_ stood out as being the major hydrocarbon, followed by H_29_, H_31_, H_25_ and other minor hydrocarbons, such as H_33_*, H_31_*, H_23_, all of them with values above one. These findings are consistent with those found in other studies [3,9,19]. Besides the saturated form, beeswax contained unsaturated structures of H_31_, H_33_ and H_35_.

The monoesters content of the beeswax samples before and after the filtration process expressed in percentage is detailed in Table 3. As it can be observed, the filtration did not change the amount of monoesters, which are important components of the beeswax, mainly saturated wax esters. The C_40_ to C_50_ monoesters with an even number of carbon atoms were quantified and the presence of an unsaturated form was found for all of them. The most abundant monoester was C_46_, followed in abundance by C_40_, C_48,_ C_44_ and C_42_. Taking into account the 100% sum of hydrocarbons and monoesters, 51.6% correspond to hydrocarbons and 48.4% correspond to monoesters. These percentages are similar to those determined by Maia & Nunes [19].

Chemical composition of beeswax is important in conferring the appropriate mechanical properties to this natural wax, especially esters of fatty acids. In addition, the alkenes, which are present in much smaller quantities than their respective alkanes, seem to have a signaling and recognition function for bees [2]. The industrial process of decontamination does not affect the basic composition of beeswax; therefore, these properties and functions will not be altered, which have special importance considering that the main use of recycled wax is to build honeycombs again.

### 2.3. Chemical Elements Composition

The presence of heavy metals and trace and ultra-trace elements in honeybees, honey or pollen is being well-studied [20,21,22] because of their use in the food industry. Furthermore, the analysis of this type of elements in beeswax is infrequent due to the fact that beeswax is not consumed; however, it is important to control the concentration of these elements because beeswax is used in other areas such as biomedicine or in the cosmetic industry. Bee products are useful tools for collecting information about environmental contamination with toxic metals [23,24,25,26]. More precisely, beeswax combs behave like a sink because of the specific lipid-based chemical composition of beeswax [2]. Trace and ultra-trace analysis of honey could be used to guarantee its place of origin [27] or to assess its quality [28].

In this particular case, the determination of chemical elements using inductively coupled plasma mass spectrometry (ICP-MS) had, as a main objective, the assessment of possible changes in the composition of wax as a consequence of the decontamination process. The results detailed in Table 4 show the difference between the chemical element composition of beeswax after subjecting it to a filtration process compared to its composition before this process. Initially, some analyzed elements, such as Cr, Cu, Mo, Ni and Sc, were absent. No shifts were found in the presence of the minerals Al, Au, Cd, Co, Cr, Cu, K, Mg, Mo, Ni, Pb, Sc, Se, V and Y. However, the concentrations of As, Ca, Fe, Hg, Mn, P, Si and Zn were altered. In particular, a significant decrease in Ca, Fe, and Zn, and a minor decrease in Hg, Mn and P were found, whereas a significant increase in As and Si was found. The largest change produced by the filtration process was found in calcium. The expected changes were a decrease in the content of chemical elements as a consequence of losses occurred during the filtration process through the bleaching earth. Nevertheless, an increase in As and Si was found. The increase in Si and As observed in the waxes after their filtration process is due to the use of diatomaceous earth as part of the filtering system. Indeed, Si represents more than 85% of diatomaceous earth composition [29], and it is possible that it is partly transferred to the filtered matrix. Furthermore, arsenic may represent around 0.2 to 0.5% of diatomaceous earth composition [29]. The transfer of arsenic that occurs during the filtration process with diatomaceous earth of alcoholic beverages has been described [30]. These authors [30] suggest the reduction of the transfer of As by selecting diatomaceous earth with low concentration of heavy metals and even pre-washing the filter before its use to filter the studied matrix. The concentrations of Si and As in the filtered waxes are within the healthy intake ranges established in the guidelines for tolerable intakes issued by the European Food Safety Authority [31].

The impact of the pesticide removal process was constrained to a low number of chemical elements: the concentrations of only eight out of twenty-three analyzed elements were modified. Regarding the main heavy metals used as indicators of environmental pollution (As, Cd, Hg and Pb), the treatment achieved the decrease in Hg. Consistent with the results obtained in other studies, concentrations of As found in pre-filtered wax and concentrations of Zn existing in post-filtered wax were similar to those found by other authors [21], whereas the other elements studied were in the same range of magnitude, although their values were different, which could be explained by the different geographical origins. Contamination with heavy metals could affect honeybee brood development, after-emergence vitality, queen productivity or longevity of adult bees [25], so the decrease in Hg is a positive effect considering the toxicity of this element for bees and human beings. The decrease in other minerals such as Mn is also beneficial because its ingestion by bees could alter brain levels of biogenic amine, and the exposure to this metal negatively affects the foraging behavior of honeybees [32]. The available information concerning the presence or the effects of chemical elements not considered environmental pollutants (Ca, P, Si, Fe, etc.) on beeswax is poor, so it is not possible to go into more details in this regard.

In view of the results obtained in the analysis of the chemical elements in the waxes, the possibility of using the chemical elements profile to determine adulteration of beeswax with other waxes was raised. Today, adulteration of beeswax with other wax can be detected very accurately because the adulteration products affect to the common concentration of beeswax components or because of the presence of no characteristic compounds of this beehive product. For this purpose, the mineral elements contents of carnauba wax, shell microcrystalline and paraffin p600, the main waxes used for beeswax adulteration, were analyzed. Figure 1 shows the mineral concentrations in the mentioned waxes. As observed, striking differences exist between waxes regarding the chemical elements profiles, especially in the case of Ca, Fe, K, Mg, P and Si, standing out as the high content of minerals in carnauba wax. The interesting differences observed in the profile of chemical elements between the different waxes analyzed raises the possibility of using this method, alone or in combination with others, to analyze possible adulterations of beeswax with carnauba wax or maybe another. In the present study, an exhaustive analysis of this possibility has not been made. Although mineral and carnauba waxes have been used to make comparisons, these have not been subjected to the filtering process, for example. It is therefore a hypothesis that needs to be developed. In fact, to be useful, it would have to allow discerning if the changes in the profile of chemical elements are due to adulteration with other waxes or to some other process used.

In addition to the chemical element profile proposed here, other aspects of the beeswax composition are used to detect adulterations. The total amount of hydrocarbons and monoesters, and the relative proportion of each of them, revealed the purity of beeswax of the tested samples [9,19,33]. Hydrocarbon is even a reliable indicator of paraffin adulteration [19].

### 2.4. Antioxidant Capacity, Polyphenol and Flavonoid Content in Unfiltered and Filtered Beeswax Samples

Total antioxidant capacity was measured using the 2,2′-azino-bis(3-ethylbenzothiazoline-6-sulphonic acid (ABTS), the ferric reducing antioxidant power (FRAP) and the 2,2-diphenyl-1-picrylhydrazyl (DPPH) assays. Total phenolic content was evaluated using the Folin-Ciocalteu’s method. Flavonoids content was quantified using a colorimetric method. Results of these three determinations conducted in unfiltered and filtered beeswax samples are shown in Table 5. These determinations were performed separately in lipophilic and hydrophilic extracts instead of in complete beeswax samples. The purpose of this separation was to avoid the underestimation of an extract. In addition, analyzing the beeswax extract first would allow the assessment of the potential utility of some of the components of beeswax in biomedical research, similar to what has already been performed with other by-products of the beeswax recycling process [34]. The total antioxidant capacity of the lipophilic extract was reduced roughly in a 45% when determined using FRAP and DPPH, while a 16% reduction was obtained when the ABTS method was used. All the results regarding antioxidant capacity of the lipophilic extract differed statistically from the results obtained before the filtration process for each technique. Similar to the total antioxidant capacity, the contents of total phenols and flavonoids were significantly reduced in 64.4% and 33.5%, respectively (*p* < 0.05). This reduction in antioxidant capacity could be a consequence of the lower content of phenolic compounds and flavonoids present in the samples after the industrial process, which could be retained by the filtration material. In contrast to the lipophilic extract, the parameters measured in the hydrophilic extract were not affected by the filtration process. However, the initial differences between both extracts are very remarkable regarding the antioxidant capacity determined using FRAP and DPPH assays and regarding the flavonoid content. It is evident, given the main lipid nature of the beeswax, that the lipophilic extract showed more antioxidant content and it was the most affected fraction after the filtration process. In view of these results, the possibility of modifying the filtration process should be considered to try to minimize the loss of antioxidants and antioxidant capacity of beeswax. Such modifications could go in several directions, such as changing the temperature or pressure conditions of the filtering process or, more likely, changing the composition or proportion of the filtering materials.

The antioxidant activity of honey could be used to classify its botanical origin [35]. However, as far as we know, this is the first time that these determinations (antioxidant capacity, total phenolics and flavonoids) have been performed in beeswax, and particularly, in its hydrophilic and lipophilic extracts. Therefore, these assessments provide a point of interest in the better characterization of beeswax and its extracts, and open the way for a potential use of some of these extracts in biomedicine.

## 3. Materials and Methods

### 3.1. Sample Collection

Beeswax samples were provided by beeswax manufacturers and private beekeepers from different regions of Spain. The samples were maintained at room temperature (20–25 °C) before being analyzed.

### 3.2. Decontamination Procedure

A total of eight old comb recycled beeswax samples were used; three for the analysis of chlorfenvinphos, three for the analysis of coumaphos and two for the analysis of *tau* fluvalinate. In all cases, the analyses were performed in duplicate. Beeswax was filtered according to a previously described method [7]. Briefly, beeswax samples were treated with the Norit SA4 PAH activated carbon (Cabot Norit America Inc., Marshall, TX, USA) and the Tonsil 114 FF diatomaceous earth (Süd-Chemie, Moosburg, Germany) in the proportion of 0.04/4.0 wt. beeswax. The filtration time ranges between 40 and 70 s. In the present study, a prototype has been used allowing to work with 500 kg of beeswax, thus scaling up the one previously described by Bonvehi & Orantes-Bermejo [7].

### 3.3. Extraction and Purification of Acaricide Residues

An amount of 0.20 g beeswax and a concentration of triphenyl phosphate (internal standard) referred to the beeswax mass (2 mg *w*/*w*) was dissolved in 3 mL of hexane in a heating block at 60 °C for three min. High-molecular weight compounds were eliminated by repeated freezing and centrifugation at 25 °C for 15 min at 3300× *g* (per triplicate). The purified solution was applied to a Florisil SPE cartridge (500 mg/3 mL) (Waters, Milford, MA, USA). The cartridges were washed with 3 mL of hexane, and the analytes were eluted with 3 mL of acetone:hexane (1:1 by volume). The eluate was evaporated to dryness under a nitrogen stream (40 °C); then the dry residue was dissolved in 2 mL of isooctane and placed in the freezer for at least 30 min [8].

### 3.4. Gas Chromatography-Mass Spectrometry (GC/MS) Analysis of Hydrocarbons and Monoesters

An amount of 120–140 mg beeswax and a concentration of eicosane (internal standard) referred to the beeswax mass 2.5–3% (*w*/*w*) were dissolved in 2.5 mL of chloroform. The solution was mechanically shaken to complete dissolution of the beeswax; then the hydrocarbons and monoesters were directly analyzed as previously described [36].

### 3.5. ICP-MS Analysis of Chemical Elements

Samples of beeswax, shell wax, p600 wax and carnauba wax were lyophilized in a vacuum pump (Telstar, Madrid, Spain) and prepared by attack with nitric acid and hydrogen peroxide of supra-pure quality in a microwave digester (Milestone, Sorisole, Italy) for the quantitative determination of metals. Determination of Mg, Al, Si, P, K, Ca, Sc, V, Cr, Mn, Fe, Co, Ni, Cu, Zn, As, Se, Y, Mo, Cd, Au, Hg and Pb total content in samples was performed using an ICP-MS instrument (Agilent 7500, Agilent Technologies, Tokyo, Japan) coupled with a Meinhard type nebulizer (Glass Expansion, Romainmotier, Switzerland) and equipped with a He collision cell. A Milli-Q system (Millipore, Bedford, MA, USA) was used to obtain deionized water (18 MΩ). All reagents used were of the highest available purity. Hydrogen peroxide and nitric acid were of supra-pure quality (Merck, Darmstadt, Germany). A standard solution of 100 µg/L of Li, Mg, Sc, Co, Y, In, Ce, Ba, Pb, Bi, and U in 1% (*v*/*v*) HNO_3_ was prepared from a 1000 mg/L multi-element stock standard solution (Merck) and used to optimize the ICP parameters daily. Single-element standard solutions for ICP-MS containing 1000 µg/mL of each analyte were also purchased from Merck. Calibration curves were prepared using Ga as an internal standard and by the dilution of stock solutions of 1000 mg/L in 1% HNO_3_. The accuracy of this method was evaluated by recovery studies after complete digestion of spiked samples with multi-element standards. The calculated recoveries for each element were between 95% and 105% in all cases.

### 3.6. Isolation of Hydrophilic and Lipophilic Extracts

Hydrophilic extraction was performed as previously reported [37] by diluting 1 g of wax in 10 mL of distilled water and filtering it through Minisart filter of 45 μm (Sigma-Aldrich Química SL, Madrid, Spain). Lipophilic extraction was performed as previously reported [38] with some modifications. Each sample (1 g) was vigorously shaken with 10 mL of n-hexane–acetone mixture (6:4) for 10 min at room temperature and filtered through Whatman No. 4 filter paper.

### 3.7. Measurement of Total Phenolic Content

The Folin–Ciocalteu method was used to determine the total phenolic content of the hydrophilic and lipophilic extracts obtained from waxes, according to the reported by Singleton et al. [39]. The hydrophilic and lipophilic extracts of each wax (0.5 mL) were mixed with 2.5 mL of 0.2 N Folin–Ciocalteu reagent for five min and 2 mL of 0.7 M sodium carbonate (Na_2_CO_3_). After incubation in the dark at 25 °C for two hours, the absorbance of the reaction mixture was measured at 760 nm using a Beckman Du 640 spectrophotometer (Instruments Inc., Fullerton, CA, USA). Gallic acid was used as standard (50–300 mg/L). The total phenolic content was expressed in mg of gallic acid equivalents/100 g (mg GAEq/100 g).

### 3.8. Measurement of Total Flavonoid Content

Flavonoid content of the hydrophilic and lipophilic extracts obtained from waxes was determined using a colorimetric method described previously by Chang et al. [40]. Briefly, a sample of 0.25 mL of the hydrophilic and lipophilic extracts of each sample or (+)-catechin standard solution was mixed with 1.25 mL of distilled water, followed by the addition of 75 μL of a 5% sodium nitrite (NaNO_2_) solution. After six min, 150 μL of a 10% cadmium chloride hemi-(pentahydrate) solution were added and allowed to settle for another five min before adding 0.5 mL of 1 M sodium hydroxide. The mixture was completed up to 2.5 mL with distilled water and mixed well. The absorbance was immediately measured against the blank (the same mixture without the sample) at 510 nm using a spectrophotometer (Beckman Du 640, USA). (+)-Catechin was used as standard (5–50 mg/L). Total flavonoid content was expressed as mg of (+)-catechin equivalents/100 g (mg CEq/100 g).

### 3.9. Quantification of the Total Antioxidant Capacity

Total antioxidant capacity of the hydrophilic and lipophilic extracts obtained from waxes was assessed using the ABTS, FRAP and the DPPH assays. The ABTS assay was performed according to the modified method of Re et al. [41]. Each sample was analyzed by triplicate and ABTS results are expressed as mmol of Trolox equivalents/kg (mmolTEq/kg). Data are reported as a mean value ± S.D. for three measurements. The FRAP assay was performed according to the protocol proposed by Benzie & Strain [42]. Three replicates of each sample were analyzed and FRAP results were expressed as mmol of Trolox equivalents/kg. The DPPH assay of the hydrophilic and lipophilic extracts obtained from waxes was performed as proposed by Brand-Williams et al. [43]. Results were expressed as µmol of Trolox equivalents/100 g (µmolTxEq/100 g).

### 3.10. Statistical Analyzes

Data are presented as mean ± SEM for a total of three determinations, unless otherwise stated. For each variable, differences between unfiltered and filtered samples were analyzed using Student’s *t* test with the statistical software SPSS 24.0 (IBM, New York, NY, USA).

## 4. Conclusions

The methodology cited here for the decontamination of waxes is a natural alternative to other existing industrial methods. It has a double advantage: (a) it does not contaminate waxes with heavy metals as a result of the process (Si and small amounts of As are incorporated into the waxes. The last could be reduced by the election of better diatomaceous earths for the filter); (b) it removes traces of environmental contaminants (heavy metals) that were initially present in the wax (Al, Cd, Pb, Mn and Hg), with the removal of Mn and the Hg being statistically significant. In this way, this method allows the revaluation of the use of beeswax in the food industry (as a fruit preservative, etc.), reverting the fall of its use in this sector caused by the presence of environmental contaminants (including heavy metals). Moreover, the composition in hydrocarbons and monoesters in the samples was not influenced by the filtration process, although this process significantly reduced the total antioxidant capacity and total phenols and flavonoids content. In addition, this study contributes to a better knowledge and characterization of beeswax, specially related to trace and ultra-trace elements and antioxidant capacity. Additionally, it offers the chance to further develop a method to effectively detect adulterations based on the chemical element profile of wax alone or in combination with other methods.

## Figures and Tables

**Figure 1 molecules-24-01518-f001:**
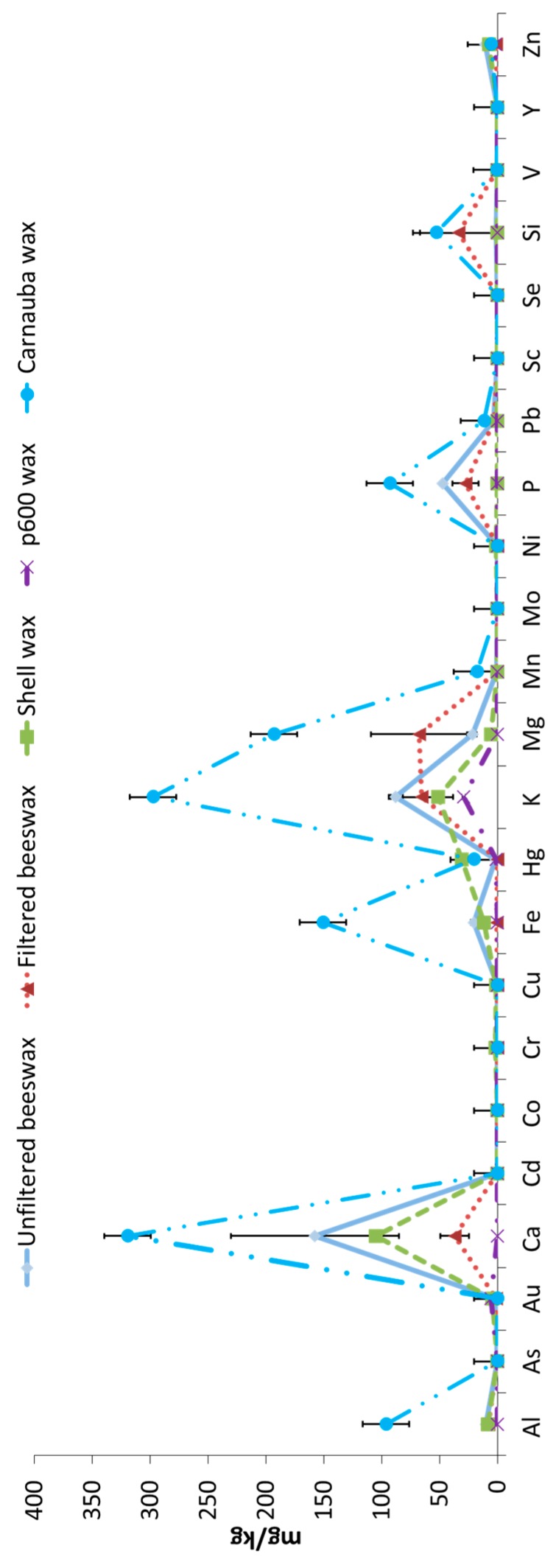
Chemical element profile of beeswax samples before and after the filtration process and of different waxes used for adulteration. Data (n = 3) shown as mean ± SEM for pre-filtered and post-filtered beeswax samples and a single measurement is shown for shell, p600 and carnauba wax.

**Table 1 molecules-24-01518-t001:** Pesticide content (ppb) and percentage reduction (%) in beeswax before and after subjecting the samples to the industrial method of decontamination.

Compounds	Initial Contamination (ppb)	Final Contamination (ppb)	Percentage Reduction
Chlorfenvinphos (n = 3)	1785 685 1398	977 59 100	45.3% (1 cycle *) 91.4% (2 cycles) 92.8% (2 cycles)
Coumaphos (n = 3)	14126 8758 1788	10914 763 58	22.8% (1 cycle) 91.3% (2 cycles) 96.7% (2 cycles)
*tau* fluvalinate ** (n = 2)	2113 2113	1721 1477	18.5% (1 cycle) 30.1% (2 cycles)

* The number of cycles (1 cycle or 2 cycles) refers to the number of times that the same sample has been subjected consecutively to the filtering process. ** In the case of *tau* fluvalinate, only two samples were analyzed after verifying that the percentage of elimination in the beeswax did not exceed 30%, similarly to Bonvehi and Orantes-Bermejo [7].

**Table 2 molecules-24-01518-t002:** Aliphatic hydrocarbons content (%) in beeswax before and after the filtration process.

Abbreviated Formula	Unfiltered Beeswax (n = 3)	Filtered Beeswax (n = 3)
	Mean ± SEM	Mean ± SEM
H19	0.143 ± 0.004	0.139 ± 0.005
H20	0.273 ± 0.083	0.485 ± 0.080
H21	0.419 ± 0.005	0.393 ± 0.019
H22	0.135 ± 0.012	0.113 ± 0.004
H23	2.979 ± 0.049	2.946 ± 0.069
H24	0.186 ± 0.002	0.169 ± 0.001
H25	5.306 ± 0.103	5.293 ± 0.077
H26	0.382 ± 0.004	0.360 ± 0.007
H27	15.473 ± 0.245	15.279 ± 0.139
H28	0.372 ± 0.020	0.334 ± 0.009
H29	9.300 ± 0.116	9.158 ± 0.215
H30	0.304 ± 0.011	0.277 ± 0.006
H31:1	2.762 ± 0.015	2.362 ± 0.262
H31	7.179 ± 0.148	7.111 ± 0.130
H32	0.351 ± 0.018	0.377 ± 0.024
H33:1	4.102 ± 0.104	4.076 ± 0.104
H33	0.737 ± 0.006	0.736 ± 0.019
H34	0.115 ± 0.004	0.109 ± 0.008
H35:1	0.213 ± 0.013	0.238 ±0.012
H35	0.418 ± 0.023	0.259 ± 0.111
H36	0.106 ± 0.053	0.185 ± 0.028
H37	0.077 ± 0.017	0.106 ± 0.010
H38	0.009 ± 0.009	0.063 ± 0.023
H39	0.049 ± 0.022	0.039 ± 0.010
H40	0.097 ± 0.005	0.118 ± 0.028
H41	0.041 ± 0.025	0.067 ± 0.018
H42	0.029 ± 0.008	0.023 ± 0.008
H43	0.043 ± 0.015	0.020 ± 0.010
H44	0.008 ± 0.004	0.012 ± 0.008
ΣH	51.609 ± 0.805	50.848 ± 0.714

ΣH represents the sum of all hydrocarbons detected.

**Table 3 molecules-24-01518-t003:** Monoesters in beeswax (%) before and after the filtration process.

Abbreviated Formula	Compound Name	Unfiltered Beeswax (n = 3)	Filtered Beeswax (n = 3)
		Mean ± SEM	Mean ± SEM
C40	Ethyl tetracontanoate	10.2 ± 0.1	10.3 ± 0.1
C40:1	Ethyl tetracontenoate	0.00 ± 0.00	0.03 ± 0.01
C42	Ethyl dotetracontanoate	5.9 ± 0.1	5.8 ± 0.1
C42:1	Ethyl dotetracontenoate	0.23 ± 0.01	0.22 ± 0.03
C44	Ethyl tetratetracontanoate	6.5 ± 0.1	6.7 ± 0.1
C44:1	Ethyl tetratetracontenoate	0.2 ± 0.1	0.2 ± 0.1
C46	Ethyl hexatetracontanoate	15.3 ± 0.9	15.4 ± 0.9
C46:1	Ethyl hexatetracontenoate	0.11 ± 0.06	0.11 ± 0.01
C48	Ethyl octatetracontanoate	8.2 ± 0.1	8.5 ± 0.1
C48:1	Ethyl octatetracontenoate	0.32 ± 0.01	0.41 ± 0.14
C50	Ethyl pentacontanoate	0.69 ± 0.08	0.65 ± 0.05
C50:1	Ethyl pentacontenoate	0.70 ± 0.11	0.73 ± 0.02
ΣC	Total monoesters	48.391 ± 0.805	49.152 ± 0.714

ΣC represents the sum of all detected monoesters.

**Table 4 molecules-24-01518-t004:** Chemical elements in unfiltered and filtered beeswax samples. Results are expressed in ppm (mg/kg).

	Unfiltered Beeswax (n = 3)	Filtered Beeswax (n = 3)
Element	Mean ± SEM	Mean ± SEM
Al	9.9 ± 2.2	7.8 ± 3.5
As	0.015 ± 0.002	0.048 ± 0.023 *
Au	1.6 ± 1.0	1.3 ± 1.1
Ca	158 ± 72	37 ± 12 *
Cd	0.020 ± 0.011	0.011 ± 0.005
Co	0.004 ± 0.004	0.000 ± 0.000
Cr	0.000 ± 0.000	0.000 ± 0.000
Cu	0.000 ± 0.000	0.000 ± 0.000
Fe	20.7 ± 2.2	0.6 ± 0.6 *
Hg	1.7 ± 1.3	0.1 ± 0.1 *
K	88 ± 6	65 ± 27
Mg	21.2 ± 3.2	67.9 ± 41.5
Mn	0.51 ± 0.17	0.14 ± 0.09 *
Mo	0.000 ± 0.000	0.000 ± 0.000
Ni	0.000 ± 0.000	0.000 ± 0.000
P	47.3 ± 1.7	27.7 ± 11.2 *
Pb	3.5 ± 2.1	2.9 ± 2.1
Sc	0.000 ± 0.000	0.000 ± 0.000
Se	0.015 ± 0.012	0.000 ± 0.000
Si	1.10 ± 1.10	33.41 ± 33.41 *
V	0.021 ± 0.004	0.044 ± 0.021
Y	0.007 ± 0.000	0.008 ± 0.002
Zn	10.3 ± 0.5	1.3 ± 0.3 *

* Statistically significant differences (*p* < 0.05) between unfiltered and filtered beeswax.

**Table 5 molecules-24-01518-t005:** Total antioxidant capacity, polyphenol and flavonoid content in unfiltered and filtered beeswax samples in the lipophilic and hydrophilic extracts (n = 3).

	Unfiltered Beeswax	Filtered Beeswax
	Mean ± SEM	Mean ± SEM
	Lipophilic extract	
ABTS (µmol TEq/Kg)	400 ± 30	330 ± 10 *
FRAP (µmol TEq/Kg)	252 ± 6	133 ± 5 *
DPPH (µmol TEq/100 g)	193 ± 29	112 ± 9 *
Total Phenolic content (mg GAEq/100 g)	0.20 ± 0.07	0.07 ± 0.03 *
Flavonoids (mg CEq/100 g)	1.2 ± 0.1	0.8 ± 0.1 *
	Hydrophilic extract	
ABTS (µmol TEq/Kg)	260 ± 10	270 ± 10
FRAP (µmol TEq/Kg)	21.3 ± 0.4	25.4 ± 1.2
DPPH (µmol TEq/100 g)	29.6 ± 11.4	66.2 ± 26.9
Total Phenolic content (mg GAEq/100 g)	0.01 ± 0.01	0.01 ± 0.01
Flavonoids (mg CEq/100 g)	0.01 ± 0.01	0.02 ± 0.01

* Statistically significant differences (*p* < 0.05) between unfiltered and filtered beeswax. CEq = Catechin equivalents. GAEq = Gallic acid equivalents. TEq = Trolox equivalents.

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
