# Peer review of "Industrial-Scale Decontamination Procedure Effects on the Content of Acaricides, Heavy Metals and Antioxidant Capacity of Beeswax"

_molecules, 2019, doi:10.3390/molecules24081518_

Round 1

Reviewer 1 Report

1.      Giving an “Abbreviations section” in front of this manuscript.

2.      Page 2, line 74: …. table 1=>.Table 1

3.      Page 4, line 99:….. H33*, H31* y H23=>.. H33*, H31*, H23

4.      Page 6, Figure 1: Resolution in Figure 1 is not enough.

5.      Page 7, Section 2.4: Please discuss how to improve the filtered beeswax process to maintain the antioxidant activity and content of phenolic compounds and flavonoids presenting in the samples after the industrial process.

6.      Page 8, line 216:…. block at 608C=>. block at 60°C; ….centrifugation at 258C for 15 min=>.centrifugation at 25°C for 15 min

7.      Page 8, line 220: …with 3 mL of acetone1hexane (111 by volume)??????

8.      Page 8, line 221:….. stream (408C) =>.. stream (40°C)

9.      Page 8, line 239:…in 1% (v/v) HNO3=>in 1% (v/v) HNO3

10.  Page 8, line 243: …in 1% (v/v) HNO3=>in 1% (v/v) HNO3

11.  Page 9, line 256: (0.5 ml) were mixed with 2.5 mL=>l or L? (consistent in unit)

12.  Page 9, line 257:….carbonate (Na2CO3) =>.carbonate (Na2CO3)

Author Response

Reviewer: Giving an “Abbreviations section” in front of this manuscript.

Authors: The authors thank the suggestion to the Reviewer. The authors have very carefully reviewed the Instructions for Authors and have not found the possibility of incorporating a section of abbreviations at the beginning of the manuscript. Point 4 of the general considerations of these Instructions indicate in relation to the abbreviations that these "should be defined in parentheses the first time they appear in the abstract, main text, and in figure or table captions and used consistently thereafter." Consequently, this is how the authors have proceeded with regard to the treatment of abbreviations throughout the manuscript. Nevertheless, authors have carefully revised the manuscript to incorporate absent abbreviations. Many thanks to the Reviewer for this appreciation.

Reviewer: Page 2, line 74: …. Table 1=>….Table 1

Authors: Done

Reviewer: Page 4, line 99:….. H33*, H31* y H23=>….. H33*, H31*, H23

Authors: Done

Reviewer: Page 6, Figure 1: Resolution in Figure 1 is not enough.

Authors: Authors have improved the resolution of Figure 1.

Reviewer: Page 7, Section 2.4: Please discuss how to improve the filtered beeswax process to maintain the antioxidant activity and content of phenolic compounds and flavonoids presenting in the samples after the industrial process.

Authors: A new paragraph has been added in order to discuss about the improvement of the filtration process in order to maintain the antioxidants and antioxidant capacity of beeswax:

Page 8, lines 328 to 334:It is evident, given the main lipid nature of the beeswax that the lipophilic extract showed more antioxidants content and it was the most affected fraction after the filtration process. In view of these results, the possibility of modifying the filtration process should be considered to try to minimize the loss of antioxidants and antioxidant capacity of beeswax. Such modifications could go in several directions, such as changing the temperature or pressure conditions of the filtering process or, more likely, changing the composition or proportion of the filtering materials.

Reviewer: Page 8, line 216:…. block at 608C=>…. block at 60°C; ….centrifugation at 258C for 15 min=>….centrifugation at 25°C for 15 min

Authors: Done

Reviewer: Page 8, line 220: …with 3 mL of acetone1hexane (111 by volume)??????

Authors: Corrected

Reviewer: Page 8, line 221:….. stream (408C) =>….. stream (40°C)

Authors: Done

Reviewer: Page 8, line 239:…in 1% (v/v) HNO3=>…in 1% (v/v) HNO3

Authors: Done

Reviewer: Page 8, line 243: …in 1% (v/v) HNO3=>…in 1% (v/v) HNO3

Authors: Done

Reviewer: Page 9, line 256: (0.5 ml) were mixed with 2.5 mL=>l or L? (consistent in unit)

Authors: Done

Reviewer: Page 9, line 257:….carbonate (Na2CO3) =>….carbonate (Na2CO

Authors: Done

Reviewer 2 Report

This paper dealing with a new method for reduces acaricides in beeswax through a filtration process. However, the manuscript has some lacks that must be reviewed before it to be considered for a new submission, including the text in English. Other points are highlighted next.

Page 1

Line 3: content of acaricides, heavy metals, and antioxidant capacity…

Line 38: Do not use words already used in the title.

Line 43: Please, exclude “in the hives”.

Line 44: Beeswax is an extremely complex…

Page 2

Line 48: Please, change “best” for “biggest”.

Line 49: Please, change “leaves” for “deposits”.

Line 50: Suggestion: “both in the apicultural and in other sectors”…

Line 51: Please, remove “areas”.

Lines 54-56: Review this paragraph.

Lines 59-62: Make the sentence clearer.

Table 1: “Pesticide content (ppb) and percentage reduction (%) in...”.

Table 1: Remove “quantified in ppb and expressed like percentage reduction.”

Table 1: Why the number of samples was different between the compounds analyzed? Why the “2 cycles” are repeated in 1st and 2nd compounds, but not in the 3rd?

Table 1: What do the cycles mean? Explain in a footnote below table 1.

Page 3

Line 93: Please, initiate the topic with text.

Table 2: …hydrocarbons content (%)…

Table 2: Remove “expressed like percentage, being the 100% sum of hydrocarbons and monoesters”.

Table 2: Does abbreviation of what? Describe the initials.

Table 2: Please, put which means * in the table.

Table 2: Include the partial sum or joint it with table 3 for shows the total sum.

Page 4

Lines 99-100: What means “values greater than
100 unity.”?

Table 3: “Monoesters in beeswax (%)...”

Table 3: Remove “expressed like percentage, being the
100% sum of hydrocarbons and monoesters.”

Page 5

Lines 129-130: Remove “are” and “expressed as mean ± SEM in mg/kg, where it is”.

Line 130: Please, change “shows” by “show”.

Line 139: What is the filtration material made of?

Table 4: Removes “composition”.

Table 4: “… samples. Results are expressed…”.

Table 4: The results pointed out that some elements had an increase after filtration. Explain better considering the filtration material used. What explains this increase?

Page 6

Line 165: This method is better applied just to determine adulterations by carnauba wax, not for all adulterants tested as proposed.

Line 166: How is it possible? The filtration process promotes expressive changes in the chemical elements profile. How knows if the changes in the elements profile in beeswax are provided by adulteration with other waxes or by any processing employed?

Further, this analysis could be made in shell wax, p600 and carnauba wax filtrated for a better understanding of the impact of each wax in the element profile.

Page 7

Table 5: Please, standardize the units of results.

Table 5: DPPH – hydrophilic extract sd is very high.

Line 204: What are the samples? Beeswax, carnauba wax, paraffin? Specify it.

Also, why the authors did not use beeswax without the application of pesticides for comparison?

Page 8

Line 208: If there were 19 samples of beeswax, why have been shown just one result for beeswax (before and after filtration)?

Were these samples mixed before filtration? Explain.

Lines 216, 218, 220, 221: Correct 608C, 258C, acetone1hexane (111…, (408C).

Page 9

The units for express total phenolic and flavonoid content as well as antioxidant activity are different from those shown in table 5.

Lines 283-284: “except if contrary is indicated”. What does it mean?

Line 288: The methodology was just cited, not presented in this paper.

Line 288: “natural and alternative”. Remove “and”.

Line 289: “to other existing industrial methods”. Which are they? They were not discussed all over the text.

Lines 289-290: “it does not contaminate waxes with heavy metals”. However, the process inputs other elements in the wax. Cite it.

Line 290: “it removes traces”. It seems that all the amount of these metals was eliminated.

Author Response

Comments and Suggestions for Authors

This paper dealing with a new method for reduces acaricides in beeswax through a filtration process. However, the manuscript has some lacks that must be reviewed before it to be considered for a new submission, including the text in English. Other points are highlighted next.

Reviewer: Page 1, line 3: content of acaricides, heavy metals, and antioxidant capacity…

Authors: At the suggestion of the Reviewer, the authors have modified the title to include "heavy metals" instead of "chemical composition".

Reviewer: Page 1, line 38: Do not use words already used in the title.

Authors: Done

Reviewer: Page 1, line 43: Please, exclude “in the hives”.

Authors: Done

Reviewer: Page 1, line 44: Beeswax is an extremely complex…

Authors: Done

Reviewer: Page 2, line 48: Please, change “best” for “biggest”.

Authors: Done

Reviewer: Page 2, line 49: Please, change “leaves” for “deposits”.

Authors: Done

Reviewer: Page 2, line 50: Suggestion: “both in the apicultural and in other sectors”…

Authors: Done

Reviewer: Page 2, line 51: Please, remove “areas”.

Authors: Done

Reviewer: Page 2, lines 54-56: Review this paragraph.

Authors: The paragraph has been modified as follows:

Page 2, lines 55 to 57:However, contamination devaluates recycled wax, causes health problems when using wax for food or other purposes, affects bee colonies, therefore, contamination has led to a gradual decline in the beeswax market in recent years.

Reviewer: Page 2, lines 59-62: Make the sentence clearer.

Authors: The authors have tried to clarify the sentence and modified it as follows:

Page 2, lines 63 to 65: “Although we must continue developing methodologies that further reduce the level of pesticides, it is also important to elucidate if any technology reducing the presence of various toxic compounds can result in changes in the chemical composition of the beeswax.

Reviewer: Table 1: “Pesticide content (ppb) and percentage reduction (%) in...”

Authors: Done

Reviewer: Table 1: Remove “quantified in ppb and expressed like percentage reduction.”

Authors: Done

Reviewer: Table 1: Why the number of samples was different between the compounds analyzed? Why the “2 cycles” are repeated in 1st and 2nd compounds, but not in the 3rd?

Authors: A footnote has been added to Table 1 in order to clarify this issue:

Page 3, Table 1:**In the case of tau fluvalinate, only two samples were analyzed after verifying that the percentage of elimination in the beeswax did not exceed 30%, similarly to Bonvehi and Orantes-Bermejo [7].

Reviewer: Table 1: What do the cycles mean? Explain in a footnote below table 1.

Authors: The number of cycles (1 cycle or 2 cycles) refers to the number of times that the same sample has been submitted consecutively to the filtering process. This has been reflected in table 1 as a footnote.

Reviewer: Page 3, line 93: Please, initiate the topic with text.

Authors: Done

Reviewer: Table 2: …hydrocarbons content (%)…

Authors: Done

Reviewer: Table 2: Remove “expressed like percentage, being the 100% sum of hydrocarbons and monoesters”.

Authors: Done

Reviewer: Table 2: Does abbreviation of what? Describe the initials.

Authors: Abbreviation refers to the chemical formula of the compound, which has been corrected by “Abbreviated formula”.

Reviewer: Table 2: Please, put which means * in the table.

Authors: The * symbols made reference to unsaturation, this has been changed in tables 2 and 3 by the correct chemical nomenclature indicating precise unsaturation degree (e.g. H31:1 instead H31*).

Reviewer: Table 2: Include the partial sum or joint it with table 3 for shows the total sum.

Authors: Done

Reviewer: Page 4 lines 99-100: What means “values greater than unity.”?

Authors: corrected by “values bigger than unit.”

Reviewer: Table 3: “Monoesters in beeswax (%)...”

Authors: Done

Reviewer: Table 3: Remove “expressed like percentage, being the
100% sum of hydrocarbons and monoesters.”

Authors: Done

Reviewer: Page 5, lines 129-130: Remove “are” and “expressed as mean ± SEM in mg/kg, where it is”.

Authors: Done

Reviewer: Page 5, line 130: Please, change “shows” by “show”.

Authors: Done

Reviewer: Page 5, line 139: What is the filtration material made of?

Authors: As stated in section 4.2, filtration material was made of active carbon and diatomaceous earth. We have indicated the materials of the filter and discuss about that:

Page 174 lines 173 to 183:The increase in Si and As observed in the waxes after their filtration process would be due to the use of diatomaceous earth as part of the filtering system. Indeed, Si represents more than 85% of diatomaceous earth composition [37], and it is possible that it is partly transferred to the filtered matrix. Furthermore, arsenic may represent around 0.2 to 0.5% of diatomaceous earth composition [37]. The transfer of arsenic have been described that occurs during the filtration process with diatomaceous earth of alcoholic beverages [38]. These authors [38] suggest the reduction of the transfer of As by selecting diatomaceous earth with low concentration of heavy metals and even pre-washings the filter before its use to filter the studied matrix. The concentrations of Si and As are within the healthy intake ranges established in the guidelines for tolerable intakes issued by the European Food Safety Authority [39].

Also, a more detailed explanation about this issue has been included in section 4.2:

Page 9 lines 350 to 353: Briefly, beeswax samples were treated with the Norit SA4 PAH activated carbon (Cabot Norit America Inc., Marshall, TX) and the Tonsil 114 FF diatomaceous earth (Süd-Chemie, Moosburg, Germany) in the proportion of 0.04/4.0 wt. beeswax

Reviewer: Table 4: Removes “composition”.

Authors: Done

Reviewer: Table 4: “… samples. Results are expressed…”.

Authors: Done

Reviewer: Table 4: The results pointed out that some elements had an increase after filtration. Explain better considering the filtration material used. What explains this increase?

Authors: The following paragraph has been added:

Page 174 lines 173 to 183:The increase in Si and As observed in the waxes after their filtration process would be due to the use of diatomaceous earth as part of the filtering system. Indeed, Si represents more than 85% of diatomaceous earth composition [37], and it is possible that it is partly transferred to the filtered matrix. Furthermore, arsenic may represent around 0.2 to 0.5% of diatomaceous earth composition [37]. The transfer of arsenic have been described that occurs during the filtration process with diatomaceous earth of alcoholic beverages [38]. These authors [38] suggest the reduction of the transfer of As by selecting diatomaceous earth with low concentration of heavy metals and even pre-washings the filter before its use to filter the studied matrix. The concentrations of Si and As are within the healthy intake ranges established in the guidelines for tolerable intakes issued by the European Food Safety Authority [39].

Reviewer: Page 6, line 165: This method is better applied just to determine adulterations by carnauba wax, not for all adulterants tested as proposed.

Authors: The sentence has been changed as follows:

Page 8, lines 295 to 298: The interesting differences observed in the profile of chemical elements between the different waxes analyzed raises the possibility of using this method, alone or in combination with others, to analyze possible adulterations of beeswax with carnauba wax or maybe other.

Reviewer: Page 6, line 166: How is it possible? The filtration process promotes expressive changes in the chemical elements profile. How knows if the changes in the elements profile in beeswax are provided by adulteration with other waxes or by any processing employed?

Further, this analysis could be made in shell wax, p600 and carnauba wax filtrated for a better understanding of the impact of each wax in the element profile.

 Authors: The authors thank and share the reflection of the Reviewer. To be more clarifying, the paragraph in question has been modified as follows:

Page 8 lines 286 to 306:In view of the results obtained in the analysis of the chemical elements in the waxes, the possibility of using the chemical elements profile to determine adulteration of beeswax with other waxes was raised. Today, adulteration of beeswax with other wax can be detected very accurately because the adulteration products affect to the common concentration of beeswax components or because of the presence of no characteristic compounds of this beeshive product. For this purpose, the mineral elements contents of carnauba wax, shell microcrystalline and paraffin p600, the main waxes used for beeswax adulteration, were analyzed. Figure 1 shows the mineral concentrations in the mentioned waxes. As can be observed, striking differences exist between waxes regarding the chemical elements profiles, especially in the case of Ca, Fe, K, Mg, P and Si, standing out the high content of minerals in carnauba wax. The interesting differences observed in the profile of chemical elements between the different waxes analyzed raises the possibility of using this method, alone or in combination with others, to analyze possible adulterations of beeswax with carnauba wax or maybe other. In the present study, an exhaustive analysis of this possibility has not been made, in fact, although mineral and carnauba waxes have been used to make comparisons, these have not been subjected to the filtering process, for example. It is therefore a hypothesis that needs to be developed. Actually, to be useful, it would have to allow discerning if the changes in the profile of chemical elements are due to adulteration with other waxes or to some process used.

Reviewer: Page 7, table 5: Please, standardize the units of results.

Authors: Done

Reviewer: Table 5: DPPH – hydrophilic extract sd is very high.

Authors: This is due to the nature of the matrix, beeswax, which is fundamentally fat-soluble. In this matrix, when the extracts are made, most of the compounds with antioxidant capacity appear in the lipophilic extract, which in general is more abundant than the hydrophilic extract. For all the parameters shown in table 5, the levels found in the lipophilic extract are much higher. In the case of the DPPH technique, the dispersion is high in the hydrophilic extract because the antioxidant values found are very low and are closer to the limit of detection of the technique than in the case of the lipophilic.

Reviewer: Page 7, line 204: What are the samples? Beeswax, carnauba wax, paraffin? Specify it.

Authors: Samples are beeswax. This has been indicated now in the text.

Reviewer: Also, why the authors did not use beeswax without the application of pesticides for comparison?

Authors:  The reviewer's assessment is very interesting. The objective of the study was to evaluate the effect of the decontamination process on various aspects related to the composition and some properties of beeswax. The objective was not to evaluate the effect of the presence of contaminants on the composition and properties of the wax. However, for the following studies the interesting suggestion of the reviewer will be taken into account.

Reviewer: Page 8, line 208: If there were 19 samples of beeswax, why have been shown just one result for beeswax (before and after filtration)? Were these samples mixed before filtration? Explain.

Authors: The authors apologize for the error relative to the number of beeswax samples used. This error was due to the fact that raw data were transcribed into the manuscript, which included the number of duplicates of each sample together with some preliminary tests. The correct number of samples used was 3 for the analysis of chlorfenvinphos, 3 for the analysis of coumaphos and 2 (for the reasons explained in the foot of Table 1) for the analysis of tau fluvalinate. In all cases, the analyzes were performed in duplicate.

The text has been modified in this sense to reflect the correct data.

Reviewer: Page 8, lines 216, 218, 220, 221: Correct 608C, 258C, acetone1hexane (111…, (408C).

Authors: Done

Reviewer: Page 9: The units for express total phenolic and flavonoid content as well as antioxidant activity are different from those shown in table 5.

Authors: The authors apologize for the confusion they may have generated by erroneously writing the units in Table 5. The correct units are those that appear in the methods section. The error has been corrected in table 5.

Reviewer: Page 9, lines 283-284: “except if contrary is indicated”. What does it mean?

Authors: Actually, this sentence should be indicated as "Unless otherwise indicated". This has been modified in the text.

Reviewer: Page 9, line 288: The methodology was just cited, not presented in this paper.

Authors: Following Reviewer suggestions, this has been changed in the text.

Reviewer: Page 9, line 288: “natural and alternative”. Remove “and”.

Authors: Done

Reviewer: Page 9, line 289: “to other existing industrial methods”. Which are they? They were not discussed all over the text.

Authors: Following the suggestion of the Reviewer, the authors have included a new sentence in the introduction section in which reference is made to other existing industrial methods and their differences with the one used in the present manuscript:

Page 2, lines 58 to 61: This new protocol sought to avoid the use of chemical agents for the bleaching of used beeswax, since these agents altered the composition of the wax and barely managed to reduce its degree of contamination [7].

Reviewer: Page 9, lines 289-290: “it does not contaminate waxes with heavy metals”. However, the process inputs other elements in the wax. Cite it.

Authors: This sentence has been changed in order to indicate that amount of Si and small amounts of As are incorporated to the waxes as consequence of the decontamination procedure:

Page 11, lines 432 to 434: a) it does not contaminate waxes with heavy metals as a result of the process (although Si and small amounts of As are incorporated into the waxes)

Reviewer: Page 9, line 290: “it removes traces”. It seems that all the amount of these metals was eliminated.

Authors: Following the Reviewer suggestion, authors changed “it removes traces” by “it diminishes the amount”.

Reviewer 3 Report

Article provide expanding knoledge on Beeswax. The applied methodology is correct. Just some consideration regarding:

Abstract section should be write in a more concise way.

Conclusion should be improved in order to highlight the novelity of this study.

Article should be accepted after typos and grammatical check and answer to request of improve the manuscript.

Author Response

Article provides expanding knowledge on Beeswax. The applied methodology is correct. Just some consideration regarding:

Reviewer: Abstract section should be write in a more concise way.

Authors: The abstract section has been revised and written in a more concise way.

Reviewer: Conclusion should be improved in order to highlight the novelty of this study.

Authors: Following Reviewer’s suggestion, authors have try to focus the conclusion.

Article should be accepted after typos and grammatical check and answer to request of improve the manuscript.

Authors: English language editing has been performed by a professional English editing service according to Reviewer’s suggestion.

Round 2

Reviewer 2 Report

The new version of the manuscript has been improved according to the suggestions made by reviewers.

Minor changes still have to make:

Line 180: The concentration of Si and As in... (beeswax, earth?) are within the healthy...

Table 5: ABTS recalculate to μmol TEq/Kg

Conclusion: incorporate As in the beeswax cannot be highlighted as an advantage. As is potentially toxic. This information should be kept, but not as an advantage. Also, it is important to highlight the importance of the quality of the filtration material used.

Author Response

ANSWERS TO REVIEWER’S QUESTIONS:

Dear Reviewer, please find attached the answer to the questions exposed by you to the last version of the manuscript. The new text introduced in this R2 version has been highlighted in green ink in order to be distinguished from modifications done in R1.

Authors really appreciate your help for the improvement of the quality of the manuscript.

Reviewer: Line 180: The concentration of Si and As in... (beeswax, earth?) are within the healthy...

Authors: Authors have modified the sentence in the following sense:

Lines 181-182:  “The concentrations of Si and As in the filtered waxes are within the healthy intake ranges established

Reviewer: Table 5: ABTS recalculate to μmol TEq/Kg

Authors: Done

Reviewer: Conclusion: incorporate As in the beeswax cannot be highlighted as an advantage. As is potentially toxic. This information should be kept, but not as an advantage. Also, it is important to highlight the importance of the quality of the filtration material used.

Authors: Authors have modified the conclusion as suggested by the Reviwer:

Lines 433-434: “as a result of the process (Si and small amounts of As are incorporated into the waxes. The last could be reduced by the election of better diatomaceous earths for the filter)